# Autonomic Phenotypes in Chronic Fatigue Syndrome (CFS) Are Associated with Illness Severity: A Cluster Analysis

**DOI:** 10.3390/jcm9082531

**Published:** 2020-08-05

**Authors:** Joanna Słomko, Fernando Estévez-López, Sławomir Kujawski, Monika Zawadka-Kunikowska, Małgorzata Tafil-Klawe, Jacek J. Klawe, Karl J. Morten, Justyna Szrajda, Modra Murovska, Julia L. Newton, Paweł Zalewski

**Affiliations:** 1Department of Hygiene, Epidemiology, Ergonomy and Postgraduate Education, Ludwik Rydygier Collegium Medicum in Bydgoszcz Nicolaus Copernicus University in Torun, M. Sklodowskiej-Curie 9, 85-094 Bydgoszcz, Poland; skujawski@cm.umk.pl (S.K.); m.zkunikowska@cm.umk.pl (M.Z.-K.); jklawe@cm.umk.pl (J.J.K.); justyna.szrajda@cm.umk.pl (J.S.); p.zalewski@cm.umk.pl (P.Z.); 2Department of Child and Adolescent Psychiatry/Psychology, Erasmus MC University Medical Center, PO Box 2040 Rotterdam, The Netherlands; fer@estevez-lopez.com; 3Department of Human Physiology, Ludwik Rydygier Collegium Medicum in Bydgoszcz Nicolaus Copernicus University in Torun, Karłowicza 24, 85-092 Bydgoszcz, Poland; malg@cm.umk.pl; 4Nuffield Department of Women’s & Reproductive Health, The Women Centre, University of Oxford, Oxford OX3 9DU, UK; karl.morten@wrh.ox.ac.uk; 5Institute of Microbiology and Virology, Riga Stradiņš University, LV-1067 Riga, Latvia; modra@latnet.lv; 6Population Health Science Institute, The Medical School, Newcastle University, Framlington Place, Newcastle-upon-Tyne NE2 4HH, UK; julia.newton@newcastle.ac.uk

**Keywords:** autonomic, chronic fatigue, quality of life

## Abstract

In this study we set out to define the characteristics of autonomic subgroups of patients with Chronic Fatigue Syndrome (CFS). The study included 131 patients with CFS (Fukuda criteria). Participants completed the following screening symptom assessment tools: Chalder Fatigue Scale, Fatigue Impact Scale, Fatigue Severity Scale, Epworth Sleepiness Scales, the self-reported Composite Autonomic Symptom Scale. Autonomic parameters were measured at rest with a Task Force Monitor (CNS Systems) and arterial stiffness using an Arteriograph (TensioMed Kft.). Principal axis factor analysis yielded four factors: fatigue, subjective and objective autonomic dysfunction and arterial stiffness. Using cluster analyses, these factors were grouped in four autonomic profiles: 34% of patients had sympathetic symptoms with dysautonomia, 5% sympathetic alone, 21% parasympathetic and 40% had issues with sympathovagal balance. Those with a sympathetic-dysautonomia phenotype were associated with more severe disease, reported greater subjective autonomic symptoms with sympathetic over-modulation and had the lowest quality of life. The highest quality of life was observed in the balance subtype where subjects were the youngest, had lower levels of fatigue and the lowest values for arterial stiffness. Future studies will aim to design autonomic profile-specific treatment interventions to determine links between autonomic phenotypes CFS and a specific treatment.

## 1. Introduction

Chronic Fatigue Syndrome (CFS) is a multi-system complex disorder, characterized by extreme mental and physical fatigue with an array of physical symptoms not relieved by rest. Additional core symptoms include post-exertional malaise, sleep disturbance, cognitive abnormalities including memory loss, poor concentration and a general reduction in cognitive ability. Over the last 20 years a variety of hypotheses have been proposed, to explain the complex etiology, including disruptions in mitochondrial, metabolic, immunological, virological, neuroendocrinological and psychological function [1]. To date there is conflicting evidence for these hypotheses with no clear evidence to support any one area as a major player. Data are emerging that biological differences are found in CFS when compared to controls. However, it remains to be proven that these are not just a consequence of inactivity and we need to start to link the biological differences to the clinical phenotype if we are to make real progress in understanding the causes of the condition. It has been suggested that deregulation of stress-responsive systems: hypothalamic-pituitary-adrenal axis, autonomic nervous system and immune system contributes to the core symptoms of CFS [2]. Autonomic imbalance has been frequently reported in patients with CFS, with symptoms including dysautonomia with orthostatic hypotonia, postural orthostatic tachycardia syndrome and gastro-intestinal disturbances being important components [3]. CSF can include sympathetic hyperactivation or parasympathetic dysfunction—fatigue, unrefreshing sleep, cognitive disturbance, post-exertional malaise could all link to this autonomic dysfunction [4].

In the present study we set out to define the characteristics of an autonomic subgroups of patients with CFS. It was hypothesized that fatigue severity was different in relation to autonomic function in patients with CFS. Identifying autonomic symptomatology in individuals with CFS will further contribute to the understanding of the role that the autonomic nervous system plays in CFS and could be useful in targeting specific treatments that will be not only effective in fatigue reduction but will be regarding to quality of life and self-perceived changes in overall health.

## 2. Experimental Section

### 2.1. Participants

The study included 131 patients with CFS who met Fukuda criteria. Participants were recruited via e-mail, telephone and mass-media advertisements. Apart from giving their written informed consent to participation in the study, the main enrolment criteria included (1) age between 25 and 65 years, (2) fatigued for more than 6 months, with a score of >36 on the Fatigue Severity Scale, (3) had at least four of additional symptoms: post-exertional malaise, headache, impaired memory and/or concentration, unrefreshing sleep, sore throat, tender lymph nodes (cervical or axillary), muscle or joint pain, (4) the fatigue must not be the result of an organic disease (i.e., psychiatric/neurological disorders (depression, anxiety, fibromyalgia, sleep disorders), neurodegenerative disorders, infectious diseases (herpes simplex virus, enterovirus, Lyme disease, Q fever, Borna disease virus), endocrine disease (hypothyroidism, diabetes mellitus, severe obesity) or immunologic disorders (lupus, multiple sclerosis, temporomandibular joint disorders)). All subjects were instructed to refrain from smoking, caffeine, alcohol ingestion and intensive physical activity on the day of investigation and ate a light breakfast only. All investigations were performed at approximately the same time of day, and took place at the chronobiology laboratory (windowless and sound-insulated room, temperature 22 °C, humidity 60%). CFS patients could participate in this study if they fulfilled the eligibility criteria for Centers for Disease Control and Prevention (Fukuda) and had been referred by a general practitioner and by the neurology and psychiatry departments. Pre-test health state assessment of subjects included: basic neurological, psychiatric, clinical examination and satisfying the above four checks. Where there was any indication of underlying illness, the patient was referred to a specialist in internal medicine, neurology or psychiatry for further investigation. An interview with a psychiatrist was scheduled if the Hospital Anxiety and Depression Scale (HADS) depression subscale score was 11 or more (to exclude a major or bipolar depressive disorder) or if the consultant suspected another psychiatric illness.

This study took place between January 2013 to July 2018 and it was approved by the Ethics Committee, Ludwik Rydygier Memorial Collegium Medicum in Bydgoszcz, Nicolaus Copernicus University, Toruń (KB 332/2013, date of approval: 25 June 2013); written informed consent was obtained from all of the participants.

### 2.2. Measures

Fatigue severity was assessed with the Chalder Fatigue Scale (CFQ), Fatigue Impact Scale (FIS) and Fatigue Severity Scale (FSS).

The Chalder Fatigue Scale consists of 11 items loading onto two dimensions of fatigue severity—mental fatigue and physical fatigue. Seven items represent physical fatigue (items 1–7) and four represent mental fatigue (items 8–11). This scale was scored in “Likert” style asking individuals 0, 1, 2 and 3 with a range from 0 to 33. Mean “Likert” score was 24.4 (SD 5.8) and 14.2 (SD 4.6) [5].

The Fatigue Impact Scale (FIS) assesses the impact that subjective fatigue has on daily functioning [6]. Forty items are each scored on a five-point Likert scale (0–4) providing a continuous scale of 0–160 with a higher score indicating greater impact [6].

The Fatigue Severity Scale (FSS) is a method of evaluating the impact of fatigue. The FSS is a short questionnaire that requires rating of the level of fatigue. The FSS questionnaire contains nine statements that rate the severity of your fatigue symptoms on a scale of 1 to 7, based on how accurately it reflects the condition during the past week. A low value (e.g., 1) indicates strong disagreement with the statement, whereas a high value (e.g., 7) indicates strong agreement. A total score of less than 36 suggests no fatigue. A total score of 36 or more suggests fatigue [7].

In view of the association between excessive daytime sleepiness and fatigue, all subjects completed the Epworth Sleepiness Scale (ESS, possible score range 0–24). This fully validated tool assesses daytime hypersomnolence, with a score ≥10 being indicative of significant hypersomnolence during the day [8].

Moreover, subjects completed the Orthostatic Grading Scale (OGS), a fully validated self-reported tool to assess the symptoms of orthostatic intolerance due to orthostatic hypotension (e.g., severity, frequency and interference with daily activities). The OGS consists of five items, each graded on a scale of 0–4; adding the scores for the individual items creates a total score of 0–20. Scores of greater or equal to 4 are considered to be consistent with orthostatic intolerance and scores greater or equal to 9 orthostatic hypotension [9].

Participants also completed the Quality of Life Scale (QOLS), which has 16 items. The QOLS is scored by adding up the score on each item to yield a total score for the instrument. Scores can range from 16 to 112. There is no automated administration or scoring software for the QOLS. Higher scores indicate a better quality of life. The average total for healthy populations is approximately 90. The QOLS has been used in studies of healthy adults and patients with rheumatic diseases, fibromyalgia, chronic obstructive pulmonary disease, gastrointestinal disorders, cardiac disease, spinal cord injury, psoriasis, urinary stress incontinence, posttraumatic stress disorder and diabetes [10].

Autonomic symptoms were measured using subjective and objective tools. First, participants completed the Autonomic Symptom Profile [11] as a self-report measure of autonomic symptoms. However, scoring was performed according to the recently abbreviated and psychometrically improved version of this questionnaire, the Composite Autonomic Symptom Score 31 (COMPASS 31) [12]. Scoring consists of 31 items from six domains—orthostatic intolerance, vasomotor, secretomotor, gastrointestinal, bladder and pupillomotor—each weighted according to the number of items and clinical relevance. Weighted individual domain scores are totaled to a maximum of 100, which indicates greater symptom load.

Moreover, autonomic parameters (Table 1) were automatically measured at rest (15 min after stabilization of the signals) with a dedicated high-tech device Task Force Monitor—TFM (CNS Systems, Gratz, Austria). The integrity of the autonomic nervous system was assessed using a three-channel ECG and continuous blood pressure monitoring (contBP with periodically cross-checked oscillometric blood pressure measurements). TFM automatically provides a power spectral analysis for heart rate variability (HRV) and blood pressure variability (BPV). HRV and BPV spectral analysis is conducted using the adaptive autoregressive model proposed by Bianchi et al. Because of the characteristics of the adaptive autoregressive model, which may produce outliers when analyzing R-wave to R-wave, all HR beat-to-beat data were filtered using Grubbs’s test for outliers’ elimination. This method of filtering is well-documented and has a strong mathematical background [13].

A high frequency band (HF) was defined as frequencies 0.15–0.4 Hz and a low frequency band (LF) as 0.04–0.15 Hz. These variability indices reflect autonomic control, with greater HF values reflecting greater vagal (parasympathetic) modulation and higher LF values indicating predominantly sympathetic modulation. Using only HRV bands when considering autonomic regulation has some limitations, therefore, TFM also provides spectral analysis of blood pressure variability, a more reliable tool for sympathetic and parasympathetic autoregulation assessment [14,15].

Arterial stiffness was measured using an Arteriograph (TensioMed Kft., Budapest, Hungary, www.tensiomed.com). This device uses a simple upper arm cuff as a sensor with the cuff pressurized to at least 35 mmHg over the actual systolic pressure. Small supra-systolic pressure changes are recorded by a high-fidelity pressure sensor in the device. In this situation the conduit arteries (subclavian, axillary, brachial) act like a cannula to transfer the central pressure changes to the edge-position sensor (similar to the central pressure measurement during cardiac catheterization). The Arteriograph first measures the actual systolic and diastolic blood pressures (BPs) oscillometrically, then the device decompresses the cuff. In a few seconds the device starts inflating the cuff again, first to the actually measured diastolic pressure, then to the suprasystolic (actually measured systolic þ35 mmHg) pressure, and records the signals for 8 s (optionally up to 10) at both cuff pressure levels. All of the signals received by the device are transmitted to a PC. The software of the device determines the augmentation index according to the manufacturer’s instructions. To determine the aortic pulse wave velocity (PWVao), the Arteriograph uses the physiological behavior of the wave reflection, namely that the ejected direct (first systolic) pulse wave is reflected back mostly from the aortic bifurcation [16,17].

### 2.3. Statistical Analyses

Analyses were performed using SPSS for Mac version 20.0 (IBM, Armonk, NY, USA) and the level of significance was set at *p* < 0.05. The fatigue, anxiety, depression, autonomic scale scores and arterial stiffness and autonomic parameters were used as input in a higher-order principal axis factor analysis with oblimin rotation to reduce the number of variables. Criteria used for excluding variables in the factor analysis were a factor loading of <0.40 or a loading of >0.32 on at least two factors. The decision about the number of factors was based on inspection of the scree plot [18] and heuristic interpretability of the factors [19]. Standardized z-scores [(value-mean)/standard deviation (SD)] were computed for the score of each participant in each variable using the mean and SD of the total sample. Internal consistency of factors was analyzed by Cronbach’s *α*.

Identification of the profiles in CFS patients consisted of two steps. First, hierarchical cluster analysis with Ward’s method was conducted to identify the optimal number of clusters (profiles), which was decided by heuristic interpretability of the clusters [20,21]. Next, a *k*-means cluster analysis was performed to allocate participants to clusters. Profiles were characterized by averaging z-scores of variables included in a same factor. Factor scores of participants with missing data were computed (averaging by the number of variables with no missing data) if their data were available at least 50% of the variables. Differences between profiles (independent variable) in categorical and continuous data (dependent variables) were analyzed by a series of *χ*^2^ test or one-way analysis of variance (ANOVA). Significance of *χ*^2^ test was based on the z-score of the difference between observed and expected frequency; a z-score >|1.96| was considered significant [22]. Within the subsample of participants who filled out the quality of life questionnaire, an independent t-test was conducted to compare the scores of the two profiles with the highest and the lowest levels of quality of life. ANOVA was not considered due to the reduced sample size of participants with scores on quality of life. Cohen’s *d* (standardized mean differences) values were interpreted as small (0.2), medium (0.5) or large (0.8) effects. Multivariate linear regression was performed to predict PWVaortic, Aixaortic and SBPaortic values (indicators of aortic stiffness) by single CFQ questions. Moreover, relationships between indicators of aortic stiffness with FIS and FSSscores were assessed. To assess the models, multivariate R2 and adjusted R2 are reported with *p*-values (denoted as Total).

## 3. Results

### 3.1. Whole Study Group Analysis

From the initial 131 interested participants, 29 were excluded as they did not meet the Fukuda criteria (*n* = 9), had an underlying psychiatric illness (*n* = 13), had another diagnosis or fatigue was not the primary complain (*n* = 7). Of the total group with CFS (*n* = 102), 66 were female (64.7%), mean age was 38.1±8.0 years and years since first episode of fatigue 4.5 ± 4.1 years. The vast majority described impaired short-term memory and concentration 90.2%, unrefreshing sleep 84.3%, post-exertional malaise 80.4%, multi-joint pain without swelling or redness 78.4%, muscle pain 61.8%, headaches 56.9%, sore throat 44.1% and tender cervical or axillary lymph nodes 28.4%. Fatigue scores assessed were 48.1 ± 8.8 using the Fatigue Severity Scale, 80.9 ± 30.0 using the Fatigue Impact Scale, 24.1 ± 4.3 using the Chalder Fatigue Scale (total), 10.5 ± 3.9 using the Chalder Fatigue Scale physical domain and 9.0 ± 1.9 using the Chalder Fatigue Scale mental domain (mean results for all individual questions of Chalder Fatigue Scale and mean results of Fatigue Severity Scale and Fatigue Impact Scale, for all participants, are available in Appendix A). Table 2 shows detailed information about the total group with CFS.

Table 3 and Table 4 show results of multivariate linear regression analysis between the mental and physical domain of the Chalder Fatigue Scale, Fatigue Severity Scale, Fatigue Impact Scale and indicators of aortic stiffness.

Multivariate linear regression analysis between single questions from Chalder Fatigue and indicators of aortic stiffness are available in Appendix A).

### 3.2. Cluster Analysis

Exploratory factor analysis was conducted to reduce the number of variables by clustering them into a reduced number of factors. Most of the assumptions for this analysis were met. However, a number of variables related to autonomic nervous system functioning showed skewed distributions (skewness >2 [22]): power spectral density (PSD) of HRV and BPV, LF-RRI, HF-RRI, LF/HF-RRI, LF/HF_RRI, HFnu-dBP, LF-dBP, HF-dBP, LF/HF-dBP, HFnu-sBP, HF-sBP and LF/HF_sBP. Data transformations did not solve this skew, and consequently, these variables were not included in the factor analysis because the factor analysis requires variables to have roughly normal score distributions. The inputs of the factor analysis are the correlations between the variables. Thus, when multicollinearity, r >|0.80| [22], between pairs of the variables were observed, one of the variables was excluded (namely, the HFnu-RRI). Factor analysis showed sampling adequacy; Kaiser-Meyer-Olkin = 0.6 [22]. The Barlett’s test of sphericity was significant (*χ*^2^ = 939.2, *p* < 0.001), which indicated that variables were (overall) significantly correlated [20]. Therefore, our factor analysis was appropriate.

The scree plot suggested a solution between two and eight factors. The four-factor solution was considered the best solution because it was the easiest to interpret by the research team (e.g., to name each of the emerging factors). The following variables were not included in the factor solution because their factor loadings were either <0.40 on all factors (i.e., the contribution to the factors was not substantial) or >0.32 on, at least, two factors (i.e., a variable substantially contributed several factors) [18]: Vasomotor (COMPASS), Bladder (COMPASS), HFnu-sBP and LF/HF-sBP (all from objectively measured autonomic nervous system). Table 5 shows the remaining 19 variables that were included in the four factors and the appropriateness of internal consistency of these factors (for all, Cronbach’s α ≥ 0.7).

Cluster analysis suggested a four-cluster solution. We named each cluster into sympathetic or parasympathetic dominant according to their sympathovagal balance during the 15 min of supine rest. This was based on previous studies and assessed using the LF/HF ratio, which was considered to suggest a sympathetic dominant pattern if LF/HF was >1 and parasympathetic if the ratio was <1 [23]. Cluster 1: Sympathetic with dysautonomia profile (LF/HF = 1.8, *n* = 35, 34%), Cluster 2: Sympathetic profile (LF/HF = 1.5, *n* = 5, 5%), Cluster 3: Parasympathetic profile (LF/HF = 0.7, *n* = 21, 21%) and Cluster 4: Balanced (LF/HF = 1, *n* = 41, 40%). Table 6 shows the characteristics of the 102 participants included in the present study (detailed information about each profile are available in Appendix B, Table A1).

Participants in the Balanced and in the Sympathetic profiles were the youngest and oldest, respectively (*p* = 0.03). There were no differences between profiles for gender, years since first episode of fatigue, education level and current job (*p*-values ranging from 0.07 for gender to 0.65 for education level). Table 7 shows that collectively the profiles were similar in Fukuda characteristics, with only one exception: post-exertional malaise more than 24 h was more common in the Sympathetic with dysautonomia (*p* = 0.02).

Figure 1 illustrates the mean factor scores at the five profiles, of which the scores are compared in Table 8.

The Sympathetic with dysautonomia profile showed high both fatigue and (subjective and objective) autonomic function and low value of arterial stiffness parameters. The Sympathetic profile was characterized by markedly low value of fatigue and subjective autonomic function, high value of objective autonomic function and markedly high value of arteriography. The Parasympathetic profile showed high values of fatigue, average subjective autonomic function, low objective autonomic function and high arterial stiffness. The Balanced profile had low values of fatigue, objective autonomic function and arteriography and average subjective autonomic function.

ANOVA showed that profiles were significantly different in quality of life; F(3, 61) = 2.98, *p* = 0.04. Figure 2 illustrates post-hoc comparisons between profiles.

Overall, our profiles reflected a continuum in adaptation to CFS. In particular, the Sympathetic with dysautonomia and Balanced profiles showed the lowest and highest quality of life, respectively; mean difference (95% Confidence interval) was 9.0 (2.8 to 15.2). Although they were not significantly different from the other profiles, quality of life seemed to be intermediate in the Parasympathetic and Sympathetic profiles. Effect sizes of the mean differences between the Sympathetic with dysautonomia and Balanced profiles were large; Cohen’s d (95% Confidence interval) was 0.8 (0.2 to 1.4).

## 4. Discussion

The autonomic nervous system may play role in many systemic diseases and there is growing evidence of autonomic dysfunction in CFS patients. The present study provided an understanding of CFS heterogeneity by including fatigue severity, autonomic symptoms and arterial stiffness measured by objective or/and subjective assessments, in a cohort of CFS patients that fulfil the Fukuda criteria. Four profiles emerged: sympathetic with dysautonomia, sympathetic, parasympathetic and balance. We found that autonomic phenotypes in CFS patients were strongly associated with measures of illness severity (measured by fatigue severity), suggesting, as there is limited evidence for the benefits of any treatment in CFS, that different approaches to treatment might be warranted. The sympathetic with dysautonomia subtype was distinguished by (1) more frequent postexertional malaise than other subtypes, (2) more severe disease expressed by high value of fatigue scales, (3) most frequently reported greater subjective autonomic symptoms with sympathetic over-modulation and (4) the lowest quality of life. Additionally, those characterized by the parasympathetic profile were at higher risk of fatigue. Patients in the sympathetic subtype were (1) the oldest, (2) at lower risk of fatigue, (3) reported the least subjective autonomic symptoms with sympathetic over-modulation and (4) had the highest value of arterial stiffness. Patients in the balance subtype were (1) the youngest, (2) at lower risk of fatigue, (3) in sympathovagal balance, (4) had the highest quality of life and (5) had the lowest value of arterial stiffness.

Previous research has reported that the most common dysautonomia symptoms in CFS are dizziness, orthostatic intolerance, palpitation, syncope, postural orthostatic tachycardia syndrome, urinary frequency and nocturia. Castro-Marrero et al. have shown that the most important autonomic symptoms in a CFS Spanish population are dizziness or cephalic instability (83.2%), episodes of orthostatic hypotension (78.8%) and motor in coordination with or without falls (76.6%) [23]. Maclachlan et al. used non-invasive autonomic measurement (Task Force^®^ Monitor) to examine whether current CFS diagnostic criteria are identifying different phenotypes of this condition. In subjects meeting different CFS criteria (Fukuda + Canadian consensus 2003, Fukuda + Canadian consensus 2003 + Canadian consensus 2011), they concluded that changes in autonomic function appear to take the form of an initial sympathetic hyperactivity followed in more severe disease by sympathetic underactivity and increased parasympathetic modulation. Moreover, the authors suggest that CFS subjects reported significantly greater autonomic impairment compared to sedentary controls. Despite that findings did not show significant objective autonomic differences, the authors raised questions about the role of co-morbid depression and sedentary controls and they indicated that more consistent inclusion and exclusion criteria across studies is necessary to understand CFS pathophysiology [24].

Our results confirm that continuous stress might result in a deregulation of autonomic nervous system manifested by sympathetic hyperactivity (sympathetic with dysautonomia profile) or sympathetic underactivity (parasympathetic profile). In our opinion, the lack of uniformity in the currently available literature regarding autonomic functioning in those with CFS may be, at least in part, related to the heterogeneity within CFS cohorts, various inclusion criteria, recruitment strategies, case ascertainment methods, degrees of psychiatric comorbidity and sociodemographic and biologic characteristics or various protocols for autonomic testing. One of our study strengths was the comprehensive nature of both the recruitment process and the characterization of the cohort using subjective and objective measures of disease severity and autonomic dysfunction. Our study also highlights the theory that abnormalities in the function of the autonomic nervous system are an important factor for better understanding CFS pathophysiology.

Other phenotypes in CFS patients have been investigated in several studies. Aslakson et al. classified CFS heterogeneity including: a polysymptomatic profile, a sore throat or painful lymph node profile, phenotypes according to the presence or absence of musculoskeletal pain [25]. Collin et al. showed that post-exertional malaise, cognitive dysfunction and unrefreshing sleep were cardinal symptoms. Muscle and joint pain, headache, sore throat and painful lymph nodes delineated three phenotypes: polysymptomatic, oligosymptomatic and pain only—these phenotypes were associated with sex, duration of illness and comorbidity [26]. Wilson et al. reported two clinically important profiles of CFS patients. Patients with profile one (68% sample) were younger, had a shorter duration of illness, lower female to male ratio, were less premorbid, had current and familial psychiatric morbidity and less functional disability. Profile two consisting of 32% of the total sample was characterized by a wider variety of symptom reports, longer duration of illness, higher female:male ratio, more functional disability, more medical visitation, more depression and anxiety [27].

Janal et al. shows that in 161 women meeting Fukuda criteria for CFS, principal components analysis of the 10 ‘minor’ symptoms of CFS produced three factors interpreted to indicate musculoskeletal, infectious and neurological subtypes. Patients in the neurological profile were at increased risk for reduced cognitive function (attention, working memory, long-term memory or rapid performance) and reduced physical ability. Those characterized by the musculoskeletal profile were at increased risk for the diagnosis of fibromyalgia and reduced physical function. Patients in the infectious subtype were at lower risk for evidence co-occurring fibromyalgia, and showed lesser risk of functional impairment [28]. Jason et al. suggest that subtyping individuals with CFS on sociodemographic (gender, age and socioeconomic status), functional disability, viral, immune, neuroendocrine, neurology, autonomic and genetic biomarkers can provide clarification for researchers and clinicians who encounter CFS’ characteristically confusing heterogeneous symptom profiles. Subgrouping is the key to understanding how CFS begins and how it is maintained, how medical and psychological variables influence its course and how it can be prevented and treated [29].

The importance of the parameters describing the arterial function (stiffness) has been shown on different groups of patients: end-stage renal disease, coronary artery disease, hypertension, diabetes and the general healthy population. A stiffer arterial system is accompanied by impaired cerebral autoregulation. Thus, hypotensive episodes occurring in the context of a stiffer arterial system may potentiate the risk of brain hypoperfusion and, consequently, of cognitive dysfunction [30,31]. There are only a limited number of studies on the effect of CFS on arterial function in this condition. Studies using pulse wave analysis have suggested that arterial stiffness may be a problem in those with CFS and could represent a marker of enhanced cardiovascular risk in this patient group [32].

In the above study, both physical and mental fatigue factors measured by CFQ scale were related to aortic stiffness. We found that arterial stiffness parameters are related to fatigue severity measured by Chalder Fatigue Scale_mental domain, Fatigue Severity Scale and Fatigue Impact Scale. Higher FIS and lower FSS scores were related to lower aortic stiffness. In a study published in 2008 [33], aortic stiffness was higher in CFS patients compared to healthy controls. In line with some results from the above study, higher elasticity of the large artery was correlated with lower subjective fatigue at rest in older woman [34]. Therefore, further studies should include Aortic stiffness assessment in CFS patients.

This study has shown that patients in the sympathetic subtype had the highest value of arterial stiffness compared to patients in the balance subtype, which had the lowest value of arterial stiffness. Arterial stiffness, an important marker of cardiovascular risk, is an independent predictor of all-cause cardiovascular morbidity and mortality. However, the impact of transient elevations in sympathetic nervous system activity on the large elastic arteries such as aorta and carotid arteries are ambiguous [35]. Swierblewska et al. showed that a chronic increase in sympathetic nervous system activity was positively correlated with higher carotid-femoral pulse wave velocity in healthy men but it was independent of age, BMI, waist circumference, waist-to-hip ratio, heart rate, pulse pressure or blood pressure [36]. In contrast, acute experimental increases in sympathetic activity in the absence of increases in BP resulted in no changes in aortic or carotid elastic compliance in young healthy humans, suggesting a rise in blood pressure is necessary to alter elastic artery stiffness [34]. Pierce suggested that additional studies are needed, but outcomes may differ based on the study protocol; namely, the method of measuring stiffness (e.g., carotid-femoral pulse wave velocity vs. local compliance), whether elevated baseline sympathetic activity is present and differences in sex, age and clinical characteristics of the participants [35]. Moreover, as one of the main underlying mechanisms of arterial stiffness is arterial fibrosis, a number of studies have focused on absolute collagen content and concentration. Collagen accumulates in the aorta with age and comorbidities such as hypertension or diabetes [37]. Alterations in the proportions of arterial collagen types, excessive intramural build-up of other proteins, such as integrins, fibronectin and desmin, may affect structural changes in the vessel wall that occur with vessel stiffening [37]. Future research might focus on the observation that collagen cross linking disorders such as Ehlers danlos and marfan syndromes are recognized in association with CFS which may influence the measurement of arterial stiffness.

One limitation of this study was the relatively small study population of CFS subjects. However, conducting clinical research in this group presents challenges in recruiting and studying a fully characterized cohort. Another limitation was that there was no specific questionnaire to measure post-exertional malaise, a core symptom of CFS. Future research is needed to determine whether the effectiveness of investigation is different across CFS phenotypes. Our next step will be to design profile-specific treatment intervention to investigate association between autonomic phenotypes in CFS patients and specific treatment outcomes in longitudinal studies.

## 5. Conclusions

The main implications of our study are: (a) identifying autonomic symptomatology in patients with CFS will contribute to the understanding of the role that the autonomic nervous system plays in CFS pathophysiology (b) subjective and/or objective autonomic function measurement should be required during the diagnostic process (c) autonomic phenotypes in CFS patients are strongly associate with measures of illness severity, suggesting that different approaches to treatment might be warranted.

## Figures and Tables

**Figure 1 jcm-09-02531-f001:**
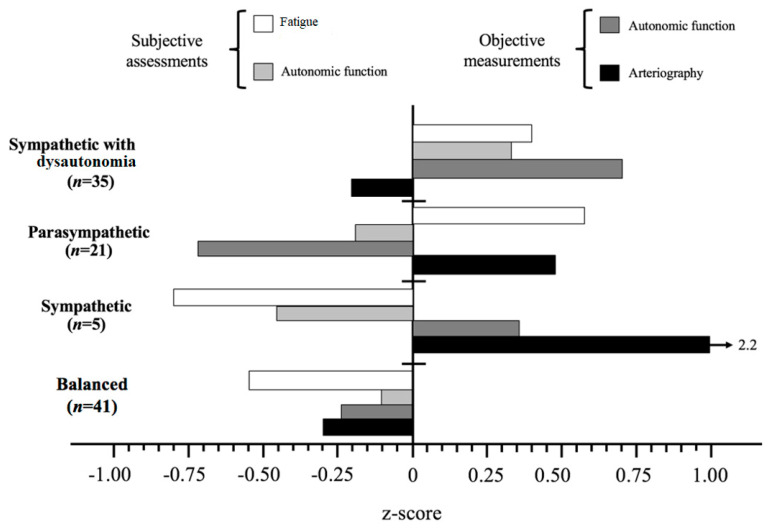
The four CFS profiles comprising four factors, *n* = 102. Post-hoc differences were tested by Student-Newman-Keuls. ANS, Autonomic Nervous System.

**Figure 2 jcm-09-02531-f002:**
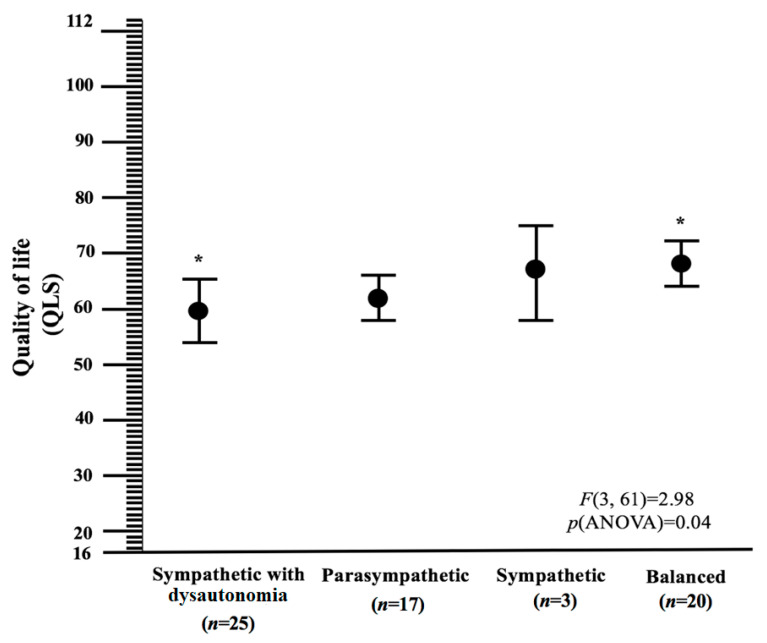
Quality of life in each profile (mean ± 95% Confidence interval, *n* = 65). One-way analysis of variance; common asterisk indicates significant differences between profiles in post-hoc comparisons. The analyses were not corrected for multiple comparisons. QLS, Quality of Life Scale (higher scores mean better quality of life).

**Table 1 jcm-09-02531-t001:** Autonomic parameters.

Sympathetic Modulation	Parasympathetic Modulation
LF-RRI [ms^2^]	HF-RRI [ms^2^]
LFnu-RRI [%]	HFnu-RRI [%]
LF-sBP [mmHg^2^]	HF-sBP [mmHg^2^]
LFnu-sBP [%]	HFnu-sBP [%]
LF-dBP [mmHg^2^]	HF-dBP [mmHg^2^]
LFnu-dBP [%]	HFnu-dBP [%]

LF, low frequency. HF, high frequency. LFnu, HFnu, frequencies calculated in normalized units. RRI, R-R interval. sBP, systolic blood pressure. dBP, diastolic blood pressure.

**Table 2 jcm-09-02531-t002:** Detailed clinical characteristics of all patients with Chronic Fatigue Syndrome (CFS).

Parameters	CFS Group	
	Mean	SD
**COMPASS 31**		
Orthostatic intolerance	11.8	11.0
Vasomotor	0.8	1.4
Secretomotor	5.5	3.9
Gastrointestinal	5.1	4.2
Bladder	0.6	0.9
Pupillomotor	1.1	1.2
CompassTotal	25.0	14.7
**Arterial stiffness**		
PWVaortic	8.4	1.7
Aixaortic	28.9	14.6
SBPaortic	130.0	17.0
**Autonomic parameters**		
LFnu-RRI	54.9	16.6
HFnu-RRI	45.1	16.6
LF-RRI	602.6	769.6
HF-RRI	520.4	731.4
LFnu-dBP	52.0	14.7
HFnu-dBP	12.9	9.8
LF-dBP	6.7	9.1
HF-dBP	1.4	2.2
LFnu-sBP	42.1	14.0
HFnu-sBP	15.5	10.3
LF-sBP	7.8	9.9
HF-sBP	2.5	3.6

**Table 3 jcm-09-02531-t003:** Relationship between results of mental and physical fatigue from the CFQ scale and arteriography in linear regression analysis.

	PWVaortic			Aixaortic			SBPaortic		
	Beta	*p*	R2	Beta	*p*	R2	Beta	*p*	R2
**CFQ_physical**	−0.06	0.56	0.4	0.05	0.65	0.4	0.08	0.43	0.4
**CFQ_mental**	−0.23	0.02	0.4	0.05	0.65	0.4	−0.08	0.44	0.4
**Total ***		0.07	0.05 (0.03)		0.84	0.00 (−0.02)		0.47	0.02 (0.00)

* Total denotes results of linear regression model (*p* value and multiple R2 (adjusted R2).

**Table 4 jcm-09-02531-t004:** Relationship between results of FSS and FIS and arterial stiffness parameters in the linear regression analysis.

	PWVaortic			Aixaortic			SBPaortic		
	Beta	*p*	R2	Beta	*p*	R2	Beta	*p*	R2
**FSS**	0.14	0.26	0.4	0.23	0.07	0.4	0.29	0.02	0.4
**FIS**	−0.33	0.01	0.4	−0.19	0.14	0.4	−0.31	0.02	0.4
**Total ***		0.03	0.07 (0.05)		0.18	0.03 (0.01)		0.04	0.07 (0.05)

* Total denotes results of linear regression model (*p* value and multiple R2 (adjusted R2).

**Table 5 jcm-09-02531-t005:** Factors emerged from higher-order principal axis factor analysis, *n* = 102.

Factor	Cronbach’s α	Scales/Measurements
Fatigue	0.86	Chalder Fatigue Scale, mental domain (CFQ), Fatigue Severity Scale and Fatigue Impact Scale
Subjective autonomic function	0.73	Chalder Fatigue Scale—physical domain (CFQ), Epworth Sleepiness Scale (ESS), Orthostatic Grading Scale (OGS, total score), orthostatic intolerance (COMPASS), secretomotor (COMPASS), gastrointestinal (COMPASS) and pupilomotor (COMPASS)
Objective autonomic function	0.68	LFnu-RRI, LFnu-dBP and LFnu-sBP (all, Task Force Monitor—TFM, CNS Systems, Gratz, Austria)
Arterial stiffness	0.76	Aortic pulse wave velocity (PWVaortic), augmentation index (Aixaortic) and central blood pressure (SBPaortic); (all, Arteriograph, TensioMed Budapest, Hungary)

Within each factor, scales/measurements are described in order of their factor loadings. CFQ, Chalder Fatigue Scale; COMPASS, Composite Autonomic Symptom Score 31; FIS, Fatigue Impact Scale; FSS, Fatigue Severity Scale; OGS, Orthostatic Grading Scale.

**Table 6 jcm-09-02531-t006:** Sociodemographic and clinical characteristics of each profile.

	Sympathetic with ANS Symptoms		Sympathetic		Parasympathetic		Balanced	
	(*n* = 35)		(*n* = 5)		(*n* = 21)		(*n* = 41)	
	Mean	SD	Mean	SD	Mean	SD	Mean	SD
**Age (years old)**	38.6	7.8	46.0	4.6	39.6	7.9	35.8	7.9
**Years since first episode of fatigue**	5.7	4.9	3.1	2.1	4.7	4.7	3.8	3.2
**Questionnaire scores**								
Chalder Fatigue Scale	25.6	4.3	18.0	3.4	26.2	3.5	21.6	3.6
Chalder Fatigue Scale_mental domain	9.9	1.5	7.2	2.5	9.8	1.2	7.9	1.9
Chalder Fatigue Scale_physical domain	11.2	4.5	7.8	2.7	9.9	3.3	7.9	1.9
Fatigue Impact Scale	94.3	23.5	34.2	21.7	99.2	23.5	58.4	25.5
Fatigue Severity Scale	51.4	6.0	39.2	7.6	54.0	6.5	40.4	10.5
	*n*	%	*n*	%	*n*	%	*n*	%
**Gender, Women, *n* (%)**	19	54.3	5	100	17	81	25	62.5
**Education level**								
Lower than secondary	1	2.9	0	0	1	4.8	2	5
Secondary	8	22.9	3	60	4	19	9	22.5
University	26	74.3	2	40	16	76.2	29	72.5
**Current job**								
Unemployed/Student	3	8.6	0	0	1	4.8	2	5
Physical work	7	20	1	20	4	19	9	22.5
Technician/Engineer	2	5.7	0	0	1	4.8	4	10
Management/Headquarters	1	2.9	0	0	5	23.8	3	7.5
Specialist/Independent profession	22	62.9	4	80	10	47.6	22	55

SD, Standard Deviation. ANS, Autonomous Nervous System.

**Table 7 jcm-09-02531-t007:** Clinical characteristics of all the participants and of each profile according to the Fukuda criteria.

	Sympathetic with Dysautonomia		Sympathetic		Parasympathetic		Balanced	
	(*n* = 35)		(*n* = 5)		(*n* = 21)		(*n* = 41)	
	*n*	%	*n*	%	*n*	%	*n*	%
Post-exertional malaise (more than 24 h)	34	97.1	4	80	14	66.7	30	73.2
Unrefreshing sleep	32	91.4	4	80	19	90.5	31	75.6
Impairment in short-term memory or concentration	35	100	4	80	18	85.7	35	85.4
Muscle pain	25	71.4	5	100	15	71.4	18	43.9
Multi-joint pain without swelling or redness	26	74.3	5	100	15	71.4	34	82.9
Headaches of a new type (pattern or severity)	20	57.1	1	20	15	71.4	22	53.7
Sore throat	21	60.0	1	20	10	47.6	13	31.7
Tender lymph nodes	10	28.6	0	0	8	38.1	11	26.8

**Table 8 jcm-09-02531-t008:** Differences in factors between subgroups, *n* = 102.

Factors ^#^	Post-hoc Differences Testing	F (3, 98)	*p*
Fatigue	Parasympathetic, Sympathetic with dysautonomia > Balanced, Sympathetic	32.57	<0.001
Subjective autonomic function	Sympathetic with dysautonomia > Balanced, Parasympathetic, Sympathetic	6.40	0.001
Objective autonomic function	Sympathetic with dysautonomia, Sympathetic > Balanced > Parasympathetic	33.57	<0.001
Arteriography	Sympathetic > Parasympathetic > Sympathetic with dysautonomia, Balanced	30.63	<0.001

^#^ Post-hoc differences were tested by Student-Newman-Keuls.

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
