# Peer review of "Autonomic Phenotypes in Chronic Fatigue Syndrome (CFS) Are Associated with Illness Severity: A Cluster Analysis"

_jcm, 2020, doi:10.3390/jcm9082531_

Round 1

Reviewer 1 Report

I have seen the revised manuscript and the comments by the authors.

I think the authors have done a good job in their responses.

The manuscript is interesting and with some novel aspects. 

Author Response

Thank you.

Reviewer 2 Report

No further comments

Author Response

Thank you.

Reviewer 3 Report

Line 394 "In study published" should be "In a study published"

This is well written and has appropriate statistics.

The paragraph starting line 391 discusses arterial stiffness. Would it be worth noting that the collagen cross linking disorders such as Ehlers danlos and marfan syndromes have an increased incidence in CFS patients and how that may influence the results.

Author Response

We thank the reviewer and are very grateful to them for giving us the opportunity to revise our manuscript. 

  1. The mistake has been corrected.
  2. We have reorganized the discussion section (line 507-514): “Moreover, as one of the main underlying mechanisms of arterial stiffness is arterial fibrosis, a number of studies have focused on absolute collagen content and concentration. Collagen accumulates in the aorta with age and comorbidities such as hypertension or diabetes [37]. Alterations in the proportions of arterial collagen types, excessive intramural build-up of other proteins, such as integrins, fibronectin, and desmin may affect structural changes in the vessel wall that occur with vessel stiffening [37]. Future research might focus on the observation that collagen cross linking disorders such as Ehlers danlos and marfan syndromes are recognized in association with CFS which may influence the measurement of arterial stiffness”.

This manuscript is a resubmission of an earlier submission. The following is a list of the peer review reports and author responses from that submission.

Round 1

Reviewer 1 Report

Autonomic dysfunction is a feature of CFS that has been repeatedly documented during the last 25 years. The orthostatic hypotension, POTS and gastrointestinal disturbances are well described, but circulatory dysregulation may also underlie many of the other characteristic symptoms, as indicated in the introduction of the manuscript.

With a lack of a reliable biomarker for CFS, with the diagnosis mainly relying on subjective description of symptoms, the patient group may be heterogeneous. It is therefore important to elucidate subgroups, and the authors wanted to define the characteristics of autonomic subgroups among patients with CFS, also how fatigue severity was related to autonomic function.

Experimental section: 

131 patients were assessed, among 29 were excluded.

The patients were included over a 5-year period (2013-2018). The Fukuda criteria, age 25-65, and score >36 on Fatigue Severity Scale  were used  for inclusion. All patients were assessed clinically for other diseases.

The experimental set-up were performed in a chronobiology laboratory.

The patients completed several self-reported symptom scales, including the Compass-31 for autonomic symptoms.

The device Task Force Monitor, with the high frequency band (HF) of heart rate variability relating to parasympathetic modulation, and the low frequency band (LF) of blood pressure variability relating to  sympathetic tone, and how these parameters reflect autonomic control is not easy to understand from the manuscript text. However, the website (CNS Systems) and references provide an overview to help the reader.

The autoregressive model and the mathematical background for filtering of signals is beyond my knowledge.

In Table 1 showing the list of autonomic parameters from the Task Force Monitor, the abbreviations should be explained, such as LF or HF -dBP (diastolic blood pressure), or LF of HF -RRI (RR interval?)

Arterial stiffness was measured using Arteriograph (TensioMed), to measure aortic pulse wave velocity, augmentation index, and central blood pressure. The abbreviations could be explained such as PWVao, which is not necessarily obvious to a reader without detailed knowledge of the specific method or research filed.

Fatigue, anxiety, depression, autonomic scale scores, arterial stiffness and autonomic parameters were used for principal axis factor analysis to determine four factors: psychological distress, subjective autonomic function, objective autonomic function, and arterial stiffness.

Then hierarchical cluster analysis was used to group the four factors in four clusters (profiles), also by assessing the LF/HF ratio.

The four autonomic profiles or phenotypes were designated as: 1: Sympathetic symptoms with dysautonomia profile; 2: Sympathetic profile; 3: Parasympathetic profile; and 4: Sympatho-vagal balance.

Then k-means cluster analysis was used to allocate study participants to these profiles.

The statistical methods used should be reviewed by an expert in statistics, also the exclusion from factor analysis of a list of variables with skewed distribution, and of variables not contributing substantially to the factors, and the conclusion that the factor analysis was appropriate. I am not able to fully evaluate these.

Differences between the four CFS profiles were then described. Characteristics of the four CFS profiles are shown in Table 4 demonstrating that especially the Sympathetic with dysautonomia group reported the highest frequency of the cardinal symptom PEM (97%). The Parasympathetic profile had highest psychological distress.

Figure 1 demonstrates the z-scores, with differences between the four CFS profiles for each of the four factors.The group Sympathetic with autonomic symptoms (n=35) had the highest score for autonomic function, also for subjective autonomic symptoms , and also with high psychological stress, but not arterial stiffness.

The sympathetic group (n=5) the highest score for arteriography, also high for objective autonomic function, but not subjective autonomic function or psychological distress. 

The parasympathetic group  (n=21) had high psychological distress and arterial stiffness, but low subjective and objective autonomic function.

The balanced group (n=41) had low values for all the four factors.

Figure 2 shows that there is a slight difference in quality of life among the four CFS groups, with the sympathetic with ANS symptoms showing the lowest, while the balanced group showed the highest mean value. What measure are the error bars (SD? 95 CI?)

This study adds knowledge of the CFS heterogeneity based on both the subjective and objective autonomic assessments, and further showed an association between the autonomic phenotypes and disease severity. Especially the group Sympathetic with dysautonomia had more frequent post-exertional malaise, more severe disease with lower quality of life, more self-reported subjective autonomic symptoms, and also the highest objective autonomic function score. 

This group also had substantial psychological distress, which could perhaps be related to a high burden of autonomic dysfunction.

In the methods and results sections the authors describe one of the factors as psychological distress, which is more common in the Parasympathetic group. Yet, in the discussion part (line 254, line 255) physiological distress is introduced. Is this a typo? or is the physiological distress meant to be a characteristics og patients with measures autonomic disturbances (which would not fit with the data in Figure 1)?

I agree that this study is important when discussing CFS patient heterogeneity, which may as the authors suggest be related to different inclusion criteria, recruitment strategies, degree of psychiatric comorbidity and biologic characteristics. The authors highlight the limitation of a relatively small study population. Thus the findings need to be reproduced in further studies.

Subgrouping is an important task to understand CFS symptom debut and maintenance, and the authors speculate on the possibility to design autonomic profile-specific treatment intervention in prospective studies.

I agree that the study strengthens the assumption that autonomic dysfunction may be an important and underlying factor for CFS pathophysiology.

The conclusions are in line with the results.

However, is it appropriate to state that autonomic phenotypes are strongly associated with measures of disease severity, taking into account Figure 2, and with a p-value from ANOVA 0,04, yet with a large effect size for difference between the Sympathetic with dysautonomia versus the Balanced profile with Cohen’s d 0,8 (the measure represented by the error bars should be stated).

The authors could perhaps discuss the data from reference 19 (Maclachlan et al, 2017). In that study the Task Force Monitor was used, and they reported higher levels of autonomic and cognitive symptoms among CFS patients than among healthy controls, however with no statistically difference in objective autonomic measures when comparing CFS patients and healthy controls. In the present study, healthy controls were not included.

Author Response

Dear Reviewer,

We would like to thank you for the constructive comments received on our manuscript. We have attempted to fully address all of the comments and have detailed our response to each and signposted where the corresponding changes have been made in the manuscript. If you have any questions or require any additional information please do not hesitate in letting me know, we are open to further improvements of our manuscript.

Reviewer's comment: Autonomic dysfunction is a feature of CFS that has been repeatedly documented during the last 25 years. The orthostatic hypotension, POTS and gastrointestinal disturbances are well described, but circulatory dysregulation may also underlie many of the other characteristic symptoms, as indicated in the introduction of the manuscript.

With a lack of a reliable biomarker for CFS, with the diagnosis mainly relying on subjective description of symptoms, the patient group may be heterogeneous. It is therefore important to elucidate subgroups, and the authors wanted to define the characteristics of autonomic subgroups among patients with CFS, also how fatigue severity was related to autonomic function.

Experimental section: 

131 patients were assessed, among 29 were excluded.

The patients were included over a 5-year period (2013-2018). The Fukuda criteria, age 25-65, and score >36 on Fatigue Severity Scale  were used  for inclusion. All patients were assessed clinically for other diseases.

The experimental set-up were performed in a chronobiology laboratory.

The patients completed several self-reported symptom scales, including the Compass-31 for autonomic symptoms.

The device Task Force Monitor, with the high frequency band (HF) of heart rate variability relating to parasympathetic modulation, and the low frequency band (LF) of blood pressure variability relating to  sympathetic tone, and how these parameters reflect autonomic control is not easy to understand from the manuscript text. However, the website (CNS Systems) and references provide an overview to help the reader.

The autoregressive model and the mathematical background for filtering of signals is beyond my knowledge.

In Table 1 showing the list of autonomic parameters from the Task Force Monitor, the abbreviations should be explained, such as LF or HF -dBP (diastolic blood pressure), or LF of HF -RRI (RR interval?) Arterial stiffness was measured using Arteriograph (TensioMed), to measure aortic pulse wave velocity, augmentation index, and central blood pressure. The abbreviations could be explained such as PWVao, which is not necessarily obvious to a reader without detailed knowledge of the specific method or research filed.

Authors' response: We apologise and have defined the abbreviations and ensured that the missing parameters are included in the methods.

Reviewer's comment: Fatigue, anxiety, depression, autonomic scale scores, arterial stiffness and autonomic parameters were used for principal axis factor analysis to determine four factors: psychological distress, subjective autonomic function, objective autonomic function, and arterial stiffness.

Then hierarchical cluster analysis was used to group the four factors in four clusters (profiles), also by assessing the LF/HF ratio.

The four autonomic profiles or phenotypes were designated as: 1: Sympathetic symptoms with dysautonomia profile; 2: Sympathetic profile; 3: Parasympathetic profile; and 4: Sympatho-vagal balance.

Then k-means cluster analysis was used to allocate study participants to these profiles.

The statistical methods used should be reviewed by an expert in statistics, also the exclusion from factor analysis of a list of variables with skewed distribution, and of variables not contributing substantially to the factors, and the conclusion that the factor analysis was appropriate. I am not able to fully evaluate these.

Differences between the four CFS profiles were then described. Characteristics of the four CFS profiles are shown in Table 4 demonstrating that especially the Sympathetic with dysautonomia group reported the highest frequency of the cardinal symptom PEM (97%). The Parasympathetic profile had highest psychological distress.

Figure 1 demonstrates the z-scores, with differences between the four CFS profiles for each of the four factors.The group Sympathetic with autonomic symptoms (n=35) had the highest score for autonomic function, also for subjective autonomic symptoms , and also with high psychological stress, but not arterial stiffness.

The sympathetic group (n=5) the highest score for arteriography, also high for objective autonomic function, but not subjective autonomic function or psychological distress. 

The parasympathetic group  (n=21) had high psychological distress and arterial stiffness, but low subjective and objective autonomic function.

The balanced group (n=41) had low values for all the four factors.

Figure 2 shows that there is a slight difference in quality of life among the four CFS groups, with the sympathetic with ANS symptoms showing the lowest, while the balanced group showed the highest mean value. What measure are the error bars (SD? 95 CI?)

Authors' response: We thank the reviewer and have added 95% CI as the measure represented by the error bars as recommend.

Reviewer's comment: This study adds knowledge of the CFS heterogeneity based on both the subjective and objective autonomic assessments, and further showed an association between the autonomic phenotypes and disease severity. Especially the group Sympathetic with dysautonomia had more frequent post-exertional malaise, more severe disease with lower quality of life, more self-reported subjective autonomic symptoms, and also the highest objective autonomic function score. 

This group also had substantial psychological distress, which could perhaps be related to a high burden of autonomic dysfunction.

In the methods and results sections the authors describe one of the factors as psychological distress, which is more common in the Parasympathetic group. Yet, in the discussion part (line 254, line 255) physiological distress is introduced. Is this a typo? or is the physiological distress meant to be a characteristics og patients with measures autonomic disturbances (which would not fit with the data in Figure 1)?

Authors' response: All mistakes has been corrected – psychological is correct.

Reviewer's comment: I agree that this study is important when discussing CFS patient heterogeneity, which may as the authors suggest be related to different inclusion criteria, recruitment strategies, degree of psychiatric comorbidity and biologic characteristics. The authors highlight the limitation of a relatively small study population. Thus the findings need to be reproduced in further studies.

Subgrouping is an important task to understand CFS symptom debut and maintenance, and the authors speculate on the possibility to design autonomic profile-specific treatment intervention in prospective studies.

I agree that the study strengthens the assumption that autonomic dysfunction may be an important and underlying factor for CFS pathophysiology.

The conclusions are in line with the results.

However, is it appropriate to state that autonomic phenotypes are strongly associated with measures of disease severity, taking into account Figure 2, and with a p-value from ANOVA 0,04, yet with a large effect size for difference between the Sympathetic with dysautonomia versus the Balanced profile with Cohen’s d 0,8 (the measure represented by the error bars should be stated).

Authors' response: We thank the reviewer and have added 95% CI as the measure represented by the error bars as recommend.

Reviewer's comment: The authors could perhaps discuss the data from reference 19 (Maclachlan et al, 2017). In that study the Task Force Monitor was used, and they reported higher levels of autonomic and cognitive symptoms among CFS patients than among healthy controls, however with no statistically difference in objective autonomic measures when comparing CFS patients and healthy controls. In the present study, healthy controls were not included.

Authors' response: We thank the reviewer and have added this as recommend.

All changes in the text are indicated in yellow background.

Reviewer 2 Report

This study of 131 clinically evaluated CFS patients meeting the 1994 research criteria documents subjective and objective measures of autonomic function and, in combination with additional symptom assessment tools, uses factor analysis to identify four subgroups with differing autonomic profiles. The findings are of interest to the field, as patient heterogeneity is a well-recognized problem. The usefulness of the study could be increased if the authors could provide additional information and clarification.

1. A full characterization of the whole study population would be helpful prior to jumping into the factor analyses. What were the scores on each of the important subscales in each questionnaire and how do these relate to normal values as well as those in other illnesses or other CFS populations. Similarly, a description of the scores and ranges for the objective measures of autonomic function and arteriography. The authors recognize that heterogeneity is a problem, and simply meeting one or more case definitions will not be enough to determine if the next groups of CFS patients are similar or different from those in this study.

2. What questionnaire was used to assess post-exertional malaise? Symptoms are listed in Table 3 but it is not clear how the information was gathered and what “yes” means. Similar, the symptoms are not clear; does malaise mean post-exertional malaise, what does “swelling” refer to? Symptoms are best queried with a scoring system that includes measures of frequency and severity.

3. In the discussion (line 245), authors state that the study provides and “understanding of CFS by including modifiable factors measured by objective and subjective assessments” – what factors are modifiable and what is intended by this statement? Are these proposed as treatment targets, if so, what evidence is there for treatment for any of the factors?

In discussion (line 249), authors state that autonomic phenotypes are strongly associated with disease severity. The results do not specify how disease severity is measured. The discussion goes on to point out differences between the groups, but strong link to severity per se is unclear. It would be of interest to provide scores for fatigue, function, and other measures of “severity” by group. For example, given that few patients are unemployed, the overall functional impairment of this study group might be less than in other studies

In discussion (line 305), authors state that the study confirmed “that the patients in the sympathetic subtype had the highest value of arterial stiffness . .” However, reference to prior report that is being confirmed is omitted.

4. Additional clarification about how the 1994 case definition is applied would be helpful. Authors state fatigue must not be the result of organic disease – what specific diagnoses are excluded and were they searched for systematically?

5. In methods, authors indicate approach used in reference 5 was used to collect patient data, but then go on to list a slightly different list of instruments. It would be helpful to reader to provide a complete list of the instruments, their origin (reference) and scoring method (perhaps in appended materials).

6. In the introduction (line 42), suggest adding the work “additional” at beginning of second sentence, so it would read “Additional core symptoms. . .” This would emphasize fatigue and associated loss of function should be present.

7. Other minor comments – definitions of abbreviations should be provided, RRI, dBP, sBP, PWVao, etc. These can be discerned from context but better to remove ambiguity. The arteriography variables PSD-RRF, PSD dBP, PSD sBP are retained in the model, but not mentioned previously in results (and not defined). The spacing in Table 2 and 5 should be improved to clarify the breaks between categories.

Author Response

Dear Reviewer,

We would like to thank you for the constructive comments received on our manuscript. We have attempted to fully address all of the comments and have detailed our response to each and signposted where the corresponding changes have been made in the manuscript. If you have any questions or require any additional information please do not hesitate in letting me know, we are open to further improvements of our manuscript.

Reviewer's comment: This study of 131 clinically evaluated CFS patients meeting the 1994 research criteria documents subjective and objective measures of autonomic function and, in combination with additional symptom assessment tools, uses factor analysis to identify four subgroups with differing autonomic profiles. The findings are of interest to the field, as patient heterogeneity is a well-recognized problem. The usefulness of the study could be increased if the authors could provide additional information and clarification.

  1. A full characterization of the whole study population would be helpful prior to jumping into the factor analyses. What were the scores on each of the important subscales in each questionnaire and how do these relate to normal values as well as those in other illnesses or other CFS populations. Similarly, a description of the scores and ranges for the objective measures of autonomic function and arteriography. The authors recognize that heterogeneity is a problem, and simply meeting one or more case definitions will not be enough to determine if the next groups of CFS patients are similar or different from those in this study.

Authors' response: Thank you for that comment. In main text we added more information about clinical characteristics of all the participants and in the appendix – full characterization of the whole study population.

Reviewer's comment: 2. What questionnaire was used to assess post-exertional malaise? Symptoms are listed in Table 3 but it is not clear how the information was gathered and what “yes” means. Similar, the symptoms are not clear; does malaise mean post-exertional malaise, what does “swelling” refer to? Symptoms are best queried with a scoring system that includes measures of frequency and severity.

Authors' response: We apologize if this table wasn’t clear enough, we have change it as ‘Clinical characteristics of all the participants and of each profile according Fukuda criteria’. Unfortunately one of the limitation of our study was not to measure PEM with specific questionnaire. We start using PEM questionnaire from 2019, and this study took place between January 2013 to July 2018.

Reviewer's comment: 3. In the discussion (line 245), authors state that the study provides and “understanding of CFS by including modifiable factors measured by objective and subjective assessments” – what factors are modifiable and what is intended by this statement? Are these proposed as treatment targets, if so, what evidence is there for treatment for any of the factors?

Authors' response: We thank the reviewer, they are correct, there is limited evidence for the benefits of any treatment in CFS. We have therefore changed the discussion to reflect this more.

Reviewer's comment: In discussion (line 249), authors state that autonomic phenotypes are strongly associated with disease severity. The results do not specify how disease severity is measured. The discussion goes on to point out differences between the groups, but strong link to severity per se is unclear. It would be of interest to provide scores for fatigue, function, and other measures of “severity” by group. For example, given that few patients are unemployed, the overall functional impairment of this study group might be less than in other studies.

Authors' response: Disease severity was measured by fatigue severity (Chalder Fatigue Scale, Fatigue Impact Scale, Fatigue Severity Scale), anxiety and depression scales. We apologize if this section wasn’t clear enough, we have reorganize this section and added more information about scores for fatigue, anxiety and depression scales scoring by group in table 3.

Reviewer's comment: In discussion (line 305), authors state that the study confirmed “that the patients in the sympathetic subtype had the highest value of arterial stiffness.” However, reference to prior report that is being confirmed is omitted.

Authors' response: Thank you for that comment, we changed it: “this study has shown that the patients in the sympathetic subtype had the highest value of arterial stiffness..”.

Reviewer's comment: 4. Additional clarification about how the 1994 case definition is applied would be helpful. Authors state fatigue must not be the result of organic disease – what specific diagnoses are excluded and were they searched for systematically?

Authors' response: We apologize if this section wasn’t clear enough, we have reorganize this section and added more detail regarding our examination of each patient and the application of the Fukuda criteria.

Reviewer's comment: 5. In methods, authors indicate approach used in reference 5 was used to collect patient data, but then go on to list a slightly different list of instruments. It would be helpful to reader to provide a complete list of the instruments, their origin (reference) and scoring method (perhaps in appended materials).

Authors' response: We thank the reviewer and have added this as recommend.

Reviewer's comment: 6. In the introduction (line 42), suggest adding the work “additional” at beginning of second sentence, so it would read “Additional core symptoms.” This would emphasize fatigue and associated loss of function should be present.

Authors' response: We thank the reviewer and have added as suggested.

Reviewer's comment: 7. Other minor comments – definitions of abbreviations should be provided, RRI, dBP, sBP, PWVao, etc. These can be discerned from context but better to remove ambiguity. The arteriography variables PSD-RRF, PSD dBP, PSD sBP are retained in the model, but not mentioned previously in results (and not defined). The spacing in Table 2 and 5 should be improved to clarify the breaks between categories.

Authors' response: We apologise and have defined the abbreviations and ensured that the missing parameters are included in the methods.

All changes in the text are indicated in yellow background.

Reviewer 3 Report

This paper is well written and uses the appropriate statistical methods. Should have sample size calculations and Bonferroni corrections applied. My major concern with this paper is to do with the interpretation of the data. Over the years many psychiatric/psychological based authors have over interpreted their data creating significant distress in the patient cohort. If one evaluates the actual responses to the questionnaires in this data you see the following: The Chalder fatigue scale is actually measuring the patients tiredness, muscle strength and concentration. The Fatigue severity scale and Fatigue impact scales are actually assessing the responses to physical fatigue (it does not segregate mental or peripheral fatigue), HADS assesses psychological responses. In the paper the scores of these are largely interpreted to be "psychological distress" when in fact most MECFS patients are reporting their post exertional malaise. Experience tells me that most patients consider they are reporting their physical fatigue scores and not their mental fatigue scores. Some of the scoring is likely to have skewing of the result data which you excluded as they did not fit normality plots and were not correctable with transformation. These data could possibly be handled using nonparametric statistical methods. I also see the use of the term Somatization disorder which is defined as the brain inducing peripheral dysfunction which is not the case in MECFS patients. The data in the paper would be best presented as question responses (possibly as supplementary data files) and not the authors interpretation of what they think it means. The real question that needs to be answered is "How does arterial stiffness relate to the actual symptoms being reported" Interpretation of the results should be balanced and give both the psychiatric and non-psychiatric interpretations. The evidence is that those with pre-existing disorders have these amplified when the patients develop MECFS.

Author Response

Dear Reviewer,

We would like to thank you for the constructive comments received on our manuscript. We have attempted to fully address all of the comments and have detailed our response to each and signposted where the corresponding changes have been made in the manuscript. If you have any questions or require any additional information please do not hesitate in letting me know, we are open to further improvements of our manuscript.

Reviewer's comment: This paper is well written and uses the appropriate statistical methods. Should have sample size calculations and Bonferroni corrections applied.

Authors' response: When the study was designed, we did not perform formal power and sample size calculations. Due to the difficulties to enrol participants who are really ME/CFS cases, we aimed at including as many participants as possible as even in the optimistic scenario we were aware that the sample would not be too large for the aims of the present study. Therefore, a limitation of the study is that the sample size was relatively small, which is acknowledge din the limitation paragraph of the discussion. Despite this limitation, this work provides a valuable contribution to the current state-of-the-art and may help to stimulate further research in larger samples.

The significant differences between the clusters Sympathethic with ANS symptoms and Balanced survives to Bonferroni adjustment. However, we would like to be honest and keep the study as it was originally designed (i.e., without adjusting for multiple comparisons). When designing the study, our rationale was that adjusting for multiple comparisons in a reduced sample size would be too stringent. Our findings may be informative in order to promote further investigations in larger samples.

Reviewer's comment: My major concern with this paper is to do with the interpretation of the data. Over the years many psychiatric/psychological based authors have over interpreted their data creating significant distress in the patient cohort. If one evaluates the actual responses to the questionnaires in this data you see the following: The Chalder fatigue scale is actually measuring the patients tiredness, muscle strength and concentration. The Fatigue severity scale and Fatigue impact scales are actually assessing the responses to physical fatigue (it does not segregate mental or peripheral fatigue), HADS assesses psychological responses. In the paper the scores of these are largely interpreted to be "psychological distress" when in fact most MECFS patients are reporting their post exertional malaise. Experience tells me that most patients consider they are reporting their physical fatigue scores and not their mental fatigue scores. Some of the scoring is likely to have skewing of the result data which you excluded as they did not fit normality plots and were not correctable with transformation. These data could possibly be handled using nonparametric statistical methods. I also see the use of the term Somatization disorder which is defined as the brain inducing peripheral dysfunction which is not the case in MECFS patients. The data in the paper would be best presented as question responses (possibly as supplementary data files) and not the authors interpretation of what they think it means. The real question that needs to be answered is "How does arterial stiffness relate to the actual symptoms being reported" Interpretation of the results should be balanced and give both the psychiatric and non-psychiatric interpretations. The evidence is that those with pre-existing disorders have these amplified when the patients develop MECFS.

Authors' response: We thank the reviewer for their comments. We understand and recognise their concerns and have redrafted the manuscript in a way that we hope they feel is more consistent with their views. Moreover, in main text and in appendix we added detailed clinical characteristics of all patients with CFS and of each profile.

We agree with the Reviewer that over the years many psychiatric/psychological authors have interpreted CFS symptoms as psychiatric disorders and it’s often was misdiagnosed as depression. That’s why in our paper one of the part of recruitment process was pre-test health state assessment with psychiatric examination. We agree with opinion that core symptoms of CFS include post-exertional malaise, but unfortunately one of the limitation of our study was not to measure PEM with specific questionnaire. We start using PEM questionnaire from 2019, and this study took place between January 2013 to July 2018.

In this paper we are trying indicate a significant importance of the disturbed function of the autonomous nervous system in the CFS etiology. In our opinion, identifying autonomic symptomatology in individuals with CFS could be useful in targeting specific treatments which will be not only effective in fatigue reduction but will be regarding to quality of life and self-perceived changes in overall health.

Changes in the text are indicated in yellow background.

Round 2

Reviewer 3 Report

This is an improvement on the 1st draft. My concern is that the paper does not separate the physical and psychological data well enough given the past injustices to the patients. The reason for saying this is that the paper as presented still allows misinterpretation of the data. I would suggest putting the mean results for all individual questions of the different questionnaires into an appendix file so that the readers can see for themselves the contribution of the psychological responses and the peripheral responses. I would suggest it may be very valuable to run regression models using the individual questions as opposed to the cumulative scores. I would put some form of evaluation of those divided into actual physical and psychological data as a paragraph in the results and discussion. One may find a higher association between arterial stiffness and individual responses vs cumulative score. Sorry to be a pest but you have important data that needs to be evaluated and presented correctly.

Line 111 a missing reference required

Line 81 you exclude fibromyalgia. This is actually an overlapping syndrome and many patients have this symptom set.

Line 215. Post exertional malaise is a cardinal defining symptom of MECFS and this shows the issues with the Fukuda criteria as they have a much wider cluster of patients than identified in other definitions.

Table 2. line 241. The scales /measures should read as the score for the various questionnaires as you have placed an interpretation of the meaning of the score in the column which may in fact be misleading as the various scales do not entirely reflect the interpretation in the table.  

Table 3. line 255. The same goes for this table. simply state the questionnaire and do not interpret it as the interpretation may be incorrect. "Fatigue, anxiety and Depression severity" should be "Questionnaire scores"

You use "physical and psychological distress" on multiple occasions. Can you come up with a more appropriate descriptor for this cluster as it can me misinterpreted by the reader??

Line 348 "one of profile" is this "one profile"??

Line 342 "as one of the core symptom of CFS" should this be "a core symptom of CFS" or some other wording?

Author Response

Reviewer's comment: This is an improvement on the 1st draft.

Authors' response: We thank the reviewer and are very grateful to them for giving us the opportunity to revise our manuscript. 

Reviewer's comment: My concern is that the paper does not separate the physical and psychological data well enough given the past injustices to the patients. The reason for saying this is that the paper as presented still allows misinterpretation of the data. I would suggest putting the mean results for all individual questions of the different questionnaires into an appendix file so that the readers can see for themselves the contribution of the psychological responses and the peripheral responses. I would suggest it may be very valuable to run regression models using the individual questions as opposed to the cumulative scores. I would put some form of evaluation of those divided into actual physical and psychological data as a paragraph in the results and discussion. One may find a higher association between arterial stiffness and individual responses vs cumulative score. Sorry to be a pest but you have important data that needs to be evaluated and presented correctly.

Authors' response: We thank the reviewer for their concerns and agree that in the past there have been issues with interpretation of data in the past. We have added physical and mental domains of Chalder Fatigue Scale. If it would make the reviewer more comfortable we would happily remove the data regarding psychological responses completely from the manuscript in order to avoid and confusion.  The inclusion of these measures is only because previous reviewers of other of our manuscripts have insisted that they be included.  If it was felt to be acceptable we would remove psychological measures.  Although we appreciate that including mean results of all individual questions could allow readers to see all of the data - this would represent an enormous table.  We would be able to do this, but are not clear whether this would be acceptable to the journal. We have added a sentence that emphasises that the individual data is available from the authors on request.  We would be grateful for guidance from the Editor how they would like us to progress with this. 

Reviewer's comment: Line 111 a missing reference required.

Authors' response: We apologise and have added this missing reference 

Reviewer's comment: Line 81 you exclude fibromyalgia. This is actually an overlapping syndrome and many patients have this symptom set.

Authors' response: We agree, however it was important to us to ensure that we defined CFS as closely as possible and as an overlapping syndrome we felt it was appropriate to exclude those with Fibro. 

Reviewer's comment: Line 215. Post exertional malaise is a cardinal defining symptom of MECFS and this shows the issues with the Fukuda criteria as they have a much wider cluster of patients than identified in other definitions.

Authors' response: We agree.

Reviewer's comment: Table 2. line 241. The scales /measures should read as the score for the various questionnaires as you have placed an interpretation of the meaning of the score in the column which may in fact be misleading as the various scales do not entirely reflect the interpretation in the table.  

Authors' response: Thank you for the observation.  We agree.

Reviewer's comment: Table 3. line 255. The same goes for this table. simply state the questionnaire and do not interpret it as the interpretation may be incorrect. "Fatigue, anxiety and Depression severity" should be "Questionnaire scores"

Authors' response: We have amended this as suggested.

Reviewer's comment: You use "physical and psychological distress" on multiple occasions. Can you come up with a more appropriate descriptor for this cluster as it can me misinterpreted by the reader??

Authors' response: We apologise and have changed this term throughout the manuscript. 

Reviewer's comment: Line 348 "one of profile" is this "one profile"??

Authors' response: We have amended. 

Reviewer's comment: Line 342 "as one of the core symptom of CFS" should this be "a core symptom of CFS" or some other wording?

Authors' response: We have amended as suggested. 

CHANGES IN THE TEXT ARE INDICATED IN GREEN BACKGROUND.